# Low-Power Highly Robust Resistance-to-Period Converter

**DOI:** 10.3390/s19010008

**Published:** 2018-12-20

**Authors:** Luis C. Álvarez-Simón, Emmanuel Gómez-Ramírez, María Teresa Sanz-Pascual

**Affiliations:** 1CONACyT—Universidad Autónoma del Estado de México, 55020 Toluca, Mexico; alvarez.simon.dr@gmail.com; 2CONACyT—Tecnológico Nacional de México—Instituto Tecnológico de la Laguna, 27000 Torreón, Mexico; 3Instituto Nacional de Astrofísica, Óptica y Electrónica, INAOE, Santa María Tonantzintla, 72840 Puebla, Mexico; materesa@inaoep.mx

**Keywords:** Resistance-to-Period converter, robust circuits, resistance measurement, readout circuit, ratiometric technique

## Abstract

This paper presents a novel structure of Resistance- to-Period (R-T) Converter highly robust to supply and temperature variations. Robustness is achieved by using the ratiometric approach so that complex circuits or high accuracy voltage references are not necessary. To prove the proposed architecture of R-T converter, a prototype was implemented in a 0.18 μm CMOS process with a single supply voltage of 1.8 V and without any stable reference voltage. Experimental results show a maximum ±1.5% output signal variation for ±10% supply voltage variation and in a 3–95 ${}^{{}^{\circ}}$C temperature range.

## 1. Introduction

In recent years, the demand for sensors has increased in many areas, from medical and consumer electronics to automotive and industrial applications. In particular, resistive sensors are widely used in the detection of several physical and chemical magnitudes, in the control of industrial processes, and in measurement and instrumentation systems [1,2,3]. Direct measurement of resistance changes can be divided into two groups depending on the range of the resistance variation: sensors whose resistance varies in several orders of magnitude, such as metal oxide gas sensors, and sensors whose relative variation in resistance is much lower than unity, such as thermistors. In the case of small resistance variations, circuits based on voltage dividers and Wheatstone bridges followed by precision differential or instrumentation amplifiers to reduce the offset voltage are used. This results in large and complex configurations and linearization techniques must be applied due to the intrinsic limitation in the dynamic range [4,5,6]. In contrast, resistance-to-frequency, -period or -duty-cycle converters are preferred if the resistance variations are very large [7,8,9,10]. This converters not only provide a wider dynamic range but also simplifies interfacing to digital systems, as no analog-to-digital converters (ADCs) are required [11,12]. In this way, the resistance is measured indirectly using a simple digital counter.

Most R-T converters proposed in the literature have mainly focused on achieving a wide dynamic range [13,14,15,16], whereas other issues such as simplicity and robustness to temperature and voltage variations have not been fully addressed, as required for low-cost, low-power integrated readout circuits for practical implementations. Accuracy and robustness of most R-T converters strongly depend on a bandgap voltage reference or on high performance building blocks, such as low-offset and high-speed comparators, resulting in higher complexity and/or power consumption [17]. In fact, the few papers on R-T converters found in the literature which consider robustness to temperature and/or voltage variations suffer from these disadvantages [18,19,20,21]. The proposed R-T converter consists of an R-I converter followed by an I-T converter previously proposed in [22]. In this paper, the ratiometric concept is applied to achieve robustness at temperature and voltage variations without increasing complexity and without the use of bandgap circuits.

The ratiometric approach is a common technique for measuring analog signals from sensors, to be conditioned in a digital signal to be compatible with a computer, DAQ system, microcontroller or microprocessor [23,24,25,26]. The advantage of the ratiometric approach is that it avoids the need for internal regulators, which reduces energy consumption and costs, but one disadvantage is that the output depends on the supply voltage, and would be a major problem where the supply voltage decreases continuously, e.g., in autonomous portable equipment. Therefore, the objective of this work was to develop an architecture robust to variations in both voltage and temperature, while also maintaining low power consumption, by using standard cells. The design of this architecture was carried out in a 0.18 μm CMOS technology and experimental results show that it is possible to achieve robustness without compromising power and area consumption.

The paper is organized as follows: after the Introduction in Section 1, the proposed approach is described in Section 2. An implementation at transistor level following the proposed approach is described in Section 3. Section 4 shows measurement results of a prototype to prove the effectiveness of the proposed approach. Finally, the conclusion is provided in Section 5.

## 2. Proposed R-T Architecture

In Figure 1, the schematic of a typical R-T converter is shown. It consists of a first block which converts the resistance value into a current, followed by a current-controlled relaxation oscillator. In the first stage, a constant voltage $V_{bias}$ provides biasing to the sensor and produces a current inversely proportional to the sensor resistance $R_{S}$, $I_{out}$ = $\frac{V_{bias}}{R_{S}}$. The robustness of this block is therefore determined by the stability of the biasing voltage $V_{bias}$.

The output current $I_{out}$ provided by the first block is driven to the relaxation oscillator, which generates a periodical signal whose period is inversely proportional to the input current and, therefore, proportional to $R_{S}$. To build the second block with simple circuits, a first-order oscillator is usually considered [27]. The input current charges and discharges a capacitor *C*, thus generating a triangular signal $V_{C}\left(t\right)$ whose slope depends on the current value $I_{out}$. $V_{C}\left(t\right)$ is compared to a reference voltage $V_{H}$ when the capacitor *C* is charged and to another reference voltage $V_{L}$ when it is discharged, being $V_{H}>V_{L}$. Each time $V_{C}\left(t\right)$ reaches $V_{H}$ or $V_{L}$, the direction of the current through the capacitor is reversed. In this way, the period of the output signal is given by:(1)$$T_{OSC}=\frac{2C(V_{H}-V_{L})}{I_{out}}$$

The value of the capacitor *C* does not practically change after fabrication. Therefore, the precision in the period of the output signal directly depends on the stability of the reference voltages.

To avoid the need for robust voltage references, the ratiometric approach [28] was applied by using the same voltage reference for both the first and the second block, i.e., $V_{bias}=\left(V_{H}-V_{L}\right)$. Thus, the period of the output signal is now given by:(2)$$T_{OSC}=2CR_{S}$$

Thus, the output of the R-T converter becomes independent of any reference voltage, and no bandgap circuit is required. This reduces complexity, cost and power consumption of the system. Furthermore, the dependence on temperature of the output signal is highly reduced, as it only depends on the capacitor temperature coefficient. When considering errors due to process, voltage and temperature (PVT) variations, the period of the output signal of the proposed R-T converter is given by:(3)$$T_{OSC}=2\left(C\pm \Delta C\right)R_{S}+T_{d}$$
where ΔC represents the deviation from the ideal capacitance value due to PVT variations and $T_{d}$ is a term that represents the delay in the comparator and switches.

The effect of the capacitor directly depends on the technology. For example, in UMC 0.18 μm CMOS technology, the variation in the MIM (Metal-Insulator-Metal) capacitance value after fabrication with respect to the nominal value is about 17.9% in the worst case. However, this error can be corrected with a calibration step, i.e., the exact value of *C* after fabrication can be determined by testing the circuit with known resistance values. Once the actual value of *C* is found, it can be used in Equation (Equation 2) to estimate $R_{S}$ from $T_{OSC}$. Note that the value of *C* remains practically constant with temperature and voltage variations because the MIM capacitor has low temperature and voltage coefficients (24 ppm/${}^{{}^{\circ}}$C and 30 ppm/V respectively).

Delay is another error source to consider. The comparison function and the change in the current direction are performed with a certain delay $t_{d}$ and, consequently, the $V_{C}\left(t\right)$ voltage surpasses $V_{H}$ and $V_{L}$, as shown in Figure 2. As can be seen, the delay $t_{d}$ is added four times in each period $T_{d}=4t_{d}$, resulting in a non-linear relationship between the sensor resistance and the period of the oscillation. To keep the output period linearly dependent on $R_{S}$, it is necessary either to reduce the delay or, if the application does not require high frequency operation, as in the case of gas resistive sensors, to choose an output $T_{OSC}$ period such that remains negligible for all the output range.

## 3. Resistance-To-Period Converter Implementation

To show the potential of the proposed architecture, this section presents a particular implementation designed in 0.18 μm CMOS technology, as shown in Figure 3. The R-I converter consists of a PMOS-input folded cascode configuration $A_{1}$ driving the MOS transistor $M_{19}$, in a series–series feedback loop. The voltage across the resistive sensor $R_{S}$ is set by the voltage follower configuration to the bias voltage $V_{bias}$. The current flowing through the sensor is driven to the Current-to-Period (I-T) converter, which is proportional to $R_{S}$. Regulated drain current mirrors are used to improve accuracy of the copy and to get a high output resistance. In this way, the current injected to or subtracted from the capacitor *C* in the I-T converter remains almost independent of the variations in $V_{C}\left(t\right)$. Amplifiers $A_{2}$ and $A_{3}$ in the regulated mirrors are a PMOS-input and an NMOS-input folded cascode amplifier, respectively. Amplifiers at transistor level Implementation are shown in Figure 4a,b, as well as its characteristics in Table 1. The bias voltages are generated without the use of bandgap circuit; the schematic at level transistors is shown in Figure 5. The circuit which controls the current flow direction in the I-T converter was presented in [22]. In this case, the switches that set the appropriate reference voltage level ($V_{H}$ or $V_{L}$) were implemented with transmission gates ($M_{24}$ to $M_{27}$) to ensure that the change in the reference level is faster than the change in the integration sign, thus avoiding a possible deadlock situation. A general purpose comparator with rail-to-rail input common-mode range was used (*Comp* in Figure 3), as shown in Figure 6 and its characteristics in Table 2.

Overall, the operation of the I-T converter is as follows: the current generated by the R-I converter is driven to the capacitor *C*. If *C* is initially being charged, when the capacitor voltage $V_{C}\left(t\right)$ surpasses the reference voltage $V_{H}$, the output of the comparator changes to the opposite state, i.e., from low to high, and so $M_{17}$ is turned off and $M_{18}$ on, and the current starts flowing out of the capacitor. At the same time, the transmission gate $M_{24}$–$M_{25}$ is turned off and $M_{26}$–$M_{27}$ on, setting the comparison level to $V_{L}$. The capacitor is now discharged so $V_{C}\left(t\right)$ decreases until it gets lower than $V_{L}$. At that moment, the output of the comparator commutes and the switches turn to their opposite state, so the charging cycle starts again.

The proposed topology uses two switches to control the sign of the integration constant and to change the reference level. Thus far, it has been supposed that both switches change their state at the same time. However, in practical applications, a deadlock situation can happen [30]. To avoid a deadlock situation, the positive feedback loop has to be stronger than the negative feedback mechanism to ensure that the transition to the next state takes place. Delay in changing the reference level estimated by simulation is of 35 ps.

In [13,16,31], two comparators and a R-S flip-flop to control the switches are used. Note that this introduces an additional error source in the period value of the output signal due to the input offsets. Therefore, high accuracy implementations to cancel this offset are necessary [32]. In contrast, when using only one comparator as in our proposed circuit, the input offset equally affects both comparison levels and its effect is cancelled, as illustrated in Figure 7 for a positive offset.

As explained in Section 2, the key of the proposal is to use the same reference voltage for the R-I and the I-T converter. In this particular implementation, a level shifter is used to convert the voltage set across the sensor, $V_{R}$, to a potential difference $V_{H}-V_{L}=V_{R}$. A pair of cascoded Flipped Voltage Followers (FVFs) [33], $M_{1}$–$M_{5}$ and $M_{6}$–$M_{10}$ in Figure 3, are used to shift the ground and $V_{R}$ voltage by the same amount. Their extremely low output resistance keeps $V_{H}$ and $V_{L}$ almost constant even during transitions in the transmission gates ($M_{24}$ to $M_{27}$). If $V_{R}$ changes, so does $V_{H}-V_{L}$ in the same proportion.

To show robustness to voltage and temperature variations, only simple reference voltages are used. For the sake of flexibility in the test of the chip, an external voltage divider was used to generate $V_{bias}$, as shown in Figure 3. Note that the divider could be integrated by replacing each resistor R by a diode-connected MOS transistor.

Analysis from the circuit using Monte Carlo simulations are shown in Figure 8. The simulation was carried out using the Monte Carlo models included in the UMC 0.18 μm Mixed-Mode 1.8 V CMOS Process, therefore all the process variables of the NMOS and PMOS transistors was into account. The standard deviation at sigma level for Monte Carlo simulation was set to Sigma = 3, recommended by the foundry. The output period versus resistance for each Monte Carlo run (index = 30) is shown in Figure 8a. The foundry establishes that, if the circuit operates correctly for all 30 iterations, there is 99% probability that over 80% of all possible component values operated correctly. To ensure the correct operation of the converter, a Monte Carlo analysis with an index equal to 100 at the worst case ($R_{S}=200$ KΩ, Temperature = 95 ${}^{{}^{\circ}}$C, $V_{DD}=1.62$ V) was made and the result is shown in Figure 8b. The Monte Carlo analysis showed a relative deviation of the Period of around 1.5% due to the process variation, taking two standard deviations around the average. This shows the robustness of the system to process variations without the use of bandgap circuits and therefore without compromising power consumption, area and complexity in the design of R-T converters using the proposed architecture.

## 4. Experimental Results

To prove the effectiveness of the proposed technique, the implementation shown in Section 3 was fabricated in 0.18 μm CMOS 1P-6M technology from UMC, with a single supply voltage of 1.8 V, where capacitor *C* has a value of 10 *p*F and the R-T converter occupies an area of 0.018 mm${}^{2}$. External pins of the realized chip are: $V_{DD}$, Gnd, $V_{R}$, $V_{bias}$, Capacitor terminal $V_{c}$ and $V_{out}$ connected by output buffers (showed in Figure 3).

Experimental measurements were carried out considering a ±10% variation in the supply voltage and an environment temperature range from 3 to 95 ${}^{{}^{\circ}}$C. For the experimental setup an E3631A Triple Output Power Supply from Keysight was used to power the test chip, a 225 MHz Universal Counter model 53132A from Agilent and an S5462SD Mixed Signal Oscilloscope from Agilent to analyze the output signal and an MTD-150 Cryotest System from Lake Shore to control the temperature of the test. The converter was tested with a variable resistance $R_{S}$ ranging from 200 kΩ to 100 MΩ. Commercial resistors were used, whose values were previously determined with a high-accuracy 2400 SourceMeter (SMU) from Keithley. The reference voltage $V_{bias}$ was established to 500 mV with a voltage divider, as explained in Section 3. The whole circuit shows a maximum power consumption of 59 μW in the worst case (when $R_{S}$ = 200 kΩ). To measure output signal, multiple-period averaging was applied. Experimental set-up is shown in Figure 9.

Figure 10a shows the fit of the experimental data to the linear Equation (Equation 3), using the weighted least square regression. The deviation of the measurement from the fitting curve, computed as the relative error ($T_{measured}-T_{fit})/T_{fit}$, is also represented. A ±1.75% linearity error was obtained for a resistance variation in the 200 kΩ–100 MΩ range. The actual value of the integrated capacitance *C*, which was found in the adjustment or calibration process, is 50.7 *p*F. Although the capacitor *C* was integrated within the chip, its top node was connected to an external pin of the chip package for test purposes. Therefore, the capacitance at the node is not only given by the MIM capacitor (*C* = 10.06 *p*F), but also by the sum of parasitic capacitances: the parasitic capacitance from the bonding wire and the capacitance of the test leads from the cryostat. Thus, the experimental data were fitted to the equation $T_{osc}$ = KT×$R_{S}$ + $T_{d}$ (where KT = 2*C*), as shown in Equation (Equation 3). After fitting, we obtained KT = 101.4 *p*F and $T_{d}$ = 1.5 μs, therefore the equivalent capacitance on the node is 50.7 *p*F. Note that, once the experimental set-up was prepared, the total capacitance remained practically constant.

When considering the variation in the output period due to ±10% variation in the supply voltage at 27 ${}^{{}^{\circ}}$C, as shown in Figure 10b, the relative error with respect to the nominal value at 1.8 V remains below ±0.7%.

In Figure 10c,d, the response and relative error of the R-T converter when exposed to supply voltage variations at the worst cases of temperature 95 ${}^{{}^{\circ}}$C and 3 ${}^{{}^{\circ}}$C, respectively, are shown. The relative error with respect to the nominal period value at 27 ${}^{{}^{\circ}}$C remains into ±1.5%.

In Table 3, the main characteristics of the proposed R-T converter are summarized. In the same table, a comparison with others CMOS R-T converters is presented. The proposed converter achieves the same robustness to temperature variations as [21] but without robust sub-circuits, which results in much lower area and power consumption. In [34], a Current-to-Frequency (C-F) converter is proposed, which shows temperature dependence and can be compared with the proposed architecture. As can be seen, the power consumption is higher, and this is because this C-F converter requires high-performance comparators to reduce their offset, which is the main source of drifts against temperature. In addition, their performance directly depends on the onboard regulator, which is not the case with the proposed R-T converter. Additionally, the proposed circuit also provides robustness to voltage variations without requiring any accurate reference voltages; in this way, effects by an AC interference superimposed on the supply voltage (in order of ±10%) could be decreased. Nevertheless, the output information has an uncertainty that limits the system resolution [35]. Compared to the other implementations shown in Table 3 that only focus on achieving a wide dynamic range, the proposed converter shows higher linearity than those of [15,36]. Although the circuit in [36] has a wider dynamic range, it makes use of high-performance blocks and, consequently, its power consumption notably increases. Experimental results demonstrate that it is possible to get a robust R-T converter without special compensation circuits, reducing the design effort and, therefore, lowering the complexity, cost and power consumption of the system.

## 5. Conclusions

In this paper, it has been proved that the ratiometric approach can be successfully used in the design of R-T converters to get robustness to supply voltage and temperature variations. The main advantage of the proposed approach is that neither accurate reference voltages nor special analog cells are required. Without an accurate biasing circuit, the implemented R-T converter presents a ±1% deviation in the period of the output signal for ±10% variation in the supply voltage and a temperature range from 3 to 95 ${}^{{}^{\circ}}$C, for almost two decades of resistance variation.

## Figures and Tables

**Figure 1 sensors-19-00008-f001:**
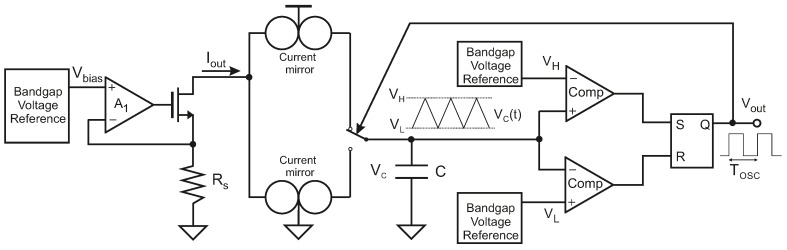
Typical resistance-to-period converter.

**Figure 2 sensors-19-00008-f002:**
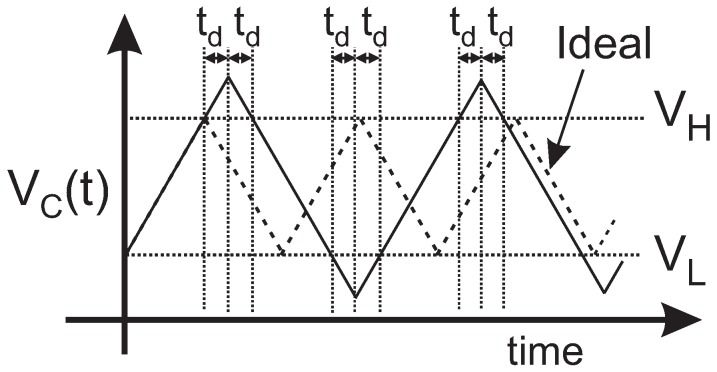
Influence of the delay in the output period.

**Figure 3 sensors-19-00008-f003:**
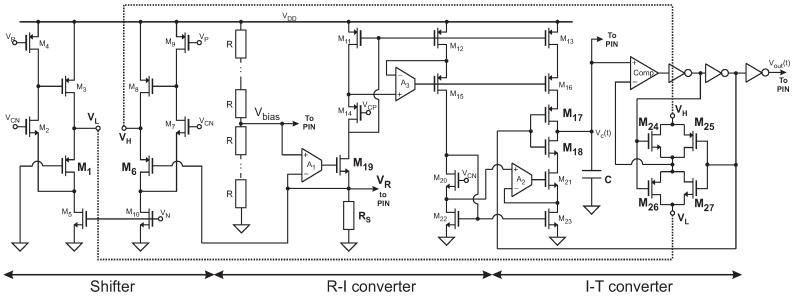
Implementation at transistor level of the proposed approach.

**Figure 4 sensors-19-00008-f004:**
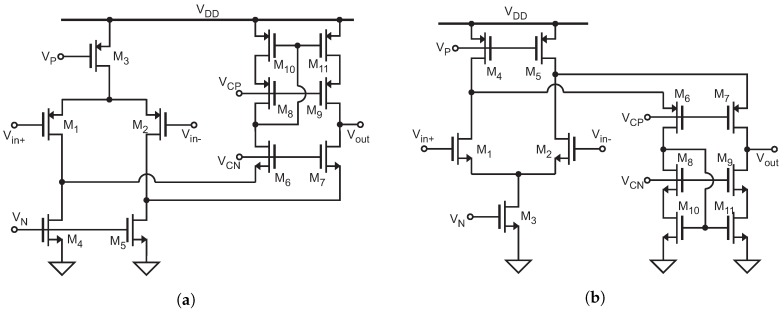
Implementation at transistor level of the Operational Transimpedance Amplifiers: (**a**) PMOS input; and (**b**) NMOS input [29].

**Figure 5 sensors-19-00008-f005:**
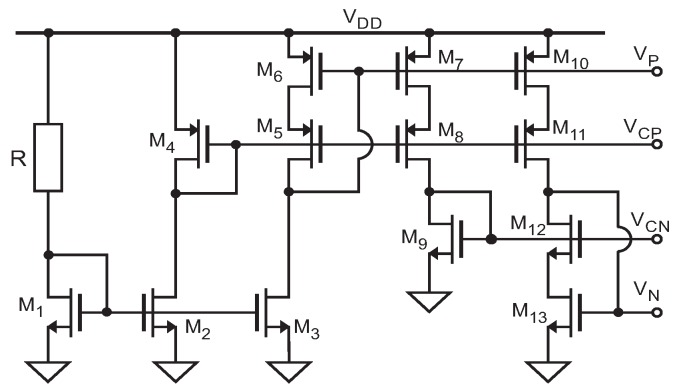
Implementation at transistor level of the bias voltages.

**Figure 6 sensors-19-00008-f006:**
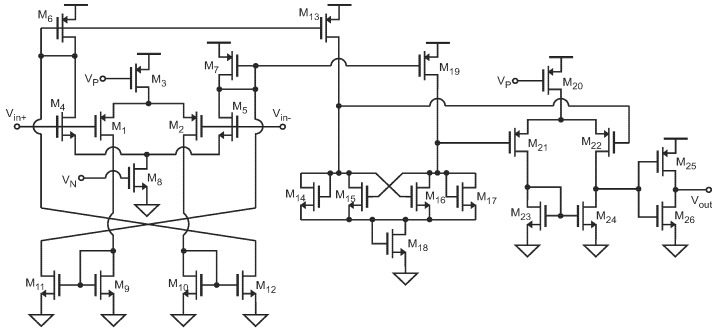
Implementation at transistor level of the comparator.

**Figure 7 sensors-19-00008-f007:**
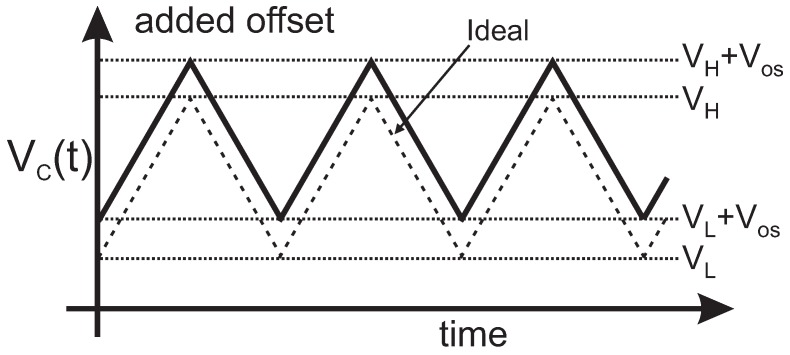
Effect of the comparator offset in the input period.

**Figure 8 sensors-19-00008-f008:**
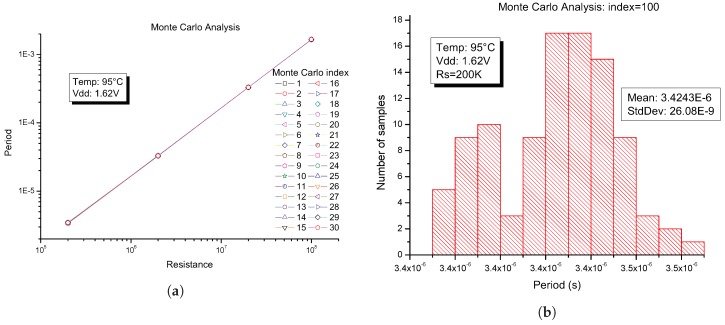
Monte Carlo analysis with sigma = 3: (**a**) output period vs. $R_{s}$; and (**b**) histogram at $R_{s}=200$ KΩ.

**Figure 9 sensors-19-00008-f009:**
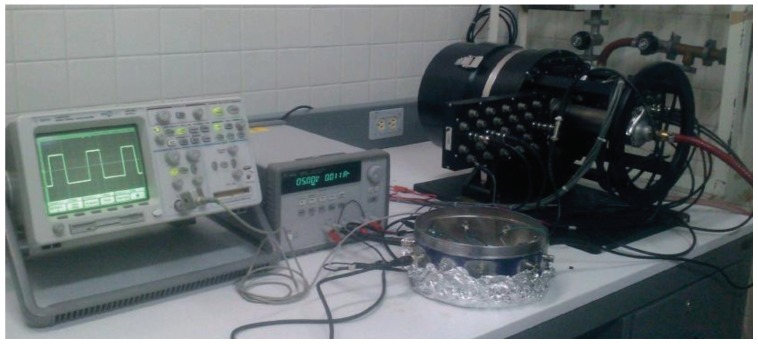
Experimental set-up.

**Figure 10 sensors-19-00008-f010:**
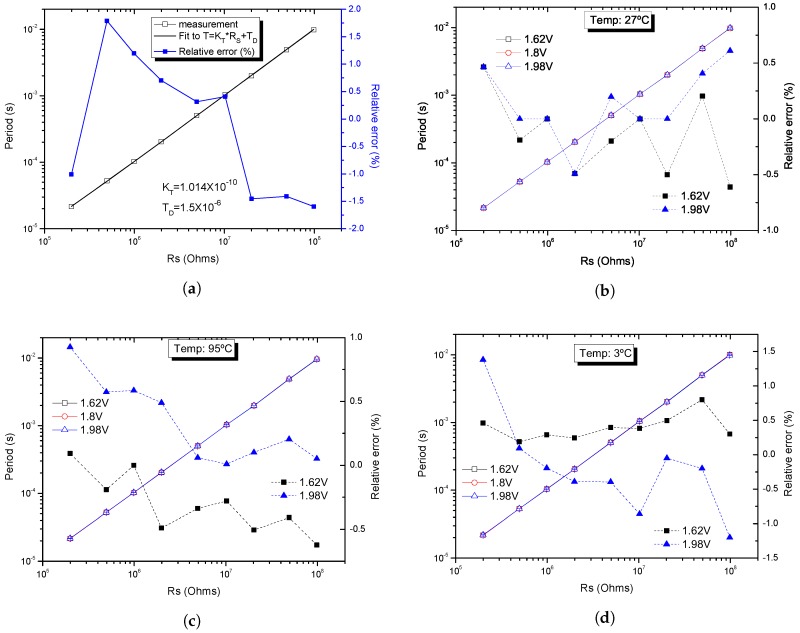
Output period vs. $R_{S}$ and: (**a**) corresponding linearity error; (**b**) corresponding relative error at different supply voltages (±10%$V_{DD}$); (**c**) corresponding relative error at different supply voltages (±10%$V_{DD}$) at 95 ${}^{{}^{\circ}}$C; and (**d**) corresponding relative error at different supply voltages (±10%$V_{DD}$) at 3 ${}^{{}^{\circ}}$C.

**Table 1 sensors-19-00008-t001:** Amplifiers parameters $A_{1}$, $A_{2}$ and $A_{3}$ used in the proposed implementation.

Parameter	NMOS Input	PMOS Input
Supply voltage	1.8 V	
DC gain	79 dB	71 dB
Phase Margin	66${}^{{}^{\circ}}$	71${}^{{}^{\circ}}$
CMRR	99 dB @20 kHz	103 dB @20 kHz
Offset	0.56 mV	0.1 mV
PSRR @dc	98 dB	71 dB
PSRR @100 kHz	77 dB	63 dB
GBW	36.4 MHz	39.4 MHz
Input reference noise	109 nV/$\sqrt{\mathrm{Hz}}$	170 nV/$\sqrt{\mathrm{Hz}}$
Corner frequency	1 kHz	
Power consumption	2.7 μW	
Silicon area	0.0002 mm${}^{2}$	0.00017 mm${}^{2}$

**Table 2 sensors-19-00008-t002:** Characteristics of comparator.

Parameter	NMOS Input
Supply voltage	1.8 V
DC gain	120 dB
Input Resolution	2 μV
Offset	0.6 mV
GBW	40.3 MHz
Propagation delay at 0.1 mV	62 ns
Propagation delay at 1 mV	51 ns
Average power consumption	17 μW
Silicon area	0.00032 mm${}^{2}$

**Table 3 sensors-19-00008-t003:** Performance summary and comparison.

Parameter	R-T Converters						
	Proposed	[15]	[21]	[37]	[38]	[34]	[36]
CMOS Technology	0.18 μm	0.13 μm	0.35 μm	0.18 μm	0.13 μm	0.65 μm	discrete components
Supply voltage (V)	1.8	1.2	3.3	1	1.8	1.2	3.3
Sensitivity	101.4 ns/kΩ	1.2 ns/kΩ	320 ns/kΩ	46 nF/10 MΩ	0.01 KHz/kΩ	41.5 Hz/nA	2.83 ns/kΩ
Input range	200 kΩ to 100 MΩ	400 Ω to 200 MΩ	470 kΩ to 100 GΩ	15 kΩ to 10 MΩ	100 Ω to 1 MΩ	30 nA to 60 μA	1 kΩ to 10 GΩ
Max. linearity error	1.75%	5%	-	-	-	±0.6 ($R^{2}$)	5%
	(2.5 decades)	(per decade)				(per decade)	(7 decades)
Max. error due to ΔTemp	±1%	-	±1%	-	-	−3.32 to +4.21%	-
Max. error due to ΔTemp and $\Delta V_{DD}$	±1.5%	-	-	-	-	-	-
Power consumption (mW)	0.06	0.32	4	0.14	0.81	0.168	25
Area (mm${}^{2}$)	0.018	-	0.84	0.175	0.125	-	-

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
