# Peer review of "Low-Power Highly Robust Resistance-to-Period Converter"

_sensors, 2018, doi:10.3390/s19010008_

Round 1

Reviewer 1 Report

Resistance to period converter uses standard idea of functional square wave generator. First of all,  state-of-the art is very short and general. It is sufficient maybe for some conference paper but definitely not for professional journal.

I recommend to show full block concept of the oscillator.

OTA design uses “folded cascode” stages (Fig. 4), it should be noted and related to some references.

GBWs of amplifiers and signal paths are not given.

I think qualitative comparison (Tab. 3) should be provided also for other similarly working types of transducers (Capacity –> period/frequency). It seems that range of measured resistance values is limited to only 2.5 decades, other works indicate better features from this point of view. However, power consumption is the best from compared solutions. But cost for this is speed most probably.

I think reference list should be extended. Not all important works seems to be noted. Some typos are at several places [10]…1596?1604, etc.

The overall quality and formal quality is very good.

Author Response

Dear Reviewer

We appreciate your time spent in reviewing our work. We have attended to each of your observations in the best way. And below you can find each of the changes that have been made.

-Resistance to period converter uses standard idea of functional square wave generator. First of all,  state-of-the art is very short and general. It is sufficient maybe for some conference paper but definitely not for professional journal.

We have increased the text of the introduction, emphasizing dependence on the power supply in quasi-digital sensors

-I recommend to show full block concept of the oscillator.

Fig. 1 was modified showing the full architecture.

-OTA design uses “folded cascode” stages (Fig. 4), it should be noted and related to some references.

Fig. 4 was referenced.

 -GBWs of amplifiers and signal paths are not given.

GBWs in Tab. 1 and 2 were added. Paths from amplifiers are connected to “bias” block shown in Fig. 5 (Fig. 5 was added).

-I think qualitative comparison (Tab. 3) should be provided also for other similarly working types of transducers (Capacity –> period/frequency). It seems that range of measured resistance values is limited to only 2.5 decades, other works indicate better features from this point of view. However, power consumption is the best from compared solutions. But cost for this is speed most probably.

An extensive comparison was made in Tab. 3. The main advantage of the proposed circuit is to get robustness to voltage and temperature variations without increasing complexity, power consumption, and bandgap circuits. A qualitative comparison was provided with reference [37] (lines 192 to 197).

-I think reference list should be extended. Not all important works seems to be noted.

Several references were added in the reference list, which were used to compare our work with others.

-Some typos are at several places [10]…1596?1604, etc.

Typos were corrected…

Best,

Authors.

Reviewer 2 Report

Alvarez-Simon et al. propose a low-power resistance-to-period converter. The paper is well written and organized. Nevertheless, there are a few concerns (see list below) related to technical content that need to be addressed before the publication.

-) In the introduction the authors should clarify the novelties of the proposed architecture with respect to that presented in their previous work cited as reference [15]

-) The possible dependence of the device behavior on the variability of the fabrication process parameters should be discuss and investigate by means of simulations and/or experimental test. For example, Monte Carlo simulation could be useful and testing of more than one chip should be performed.

Author Response

Dear Reviewer

We appreciate your time spent in reviewing our work. We have attended to each of your observations in the best way. And below you can find each of the changes that have been made.

- In the introduction the authors should clarify the novelties of the proposed architecture with respect to that presented in their previous work cited as reference [15]

We have increased the text of the introduction, emphasizing the difference between our previous work with actual work (lines 33 to 36 and 42 to 46).

- The possible dependence of the device behavior on the variability of the fabrication process parameters should be discuss and investigate by means of simulations and/or experimental test. For example, Monte Carlo simulation could be useful and testing of more than one chip should be performed.

We added Fig 8. where a Monte Carlo simulations were done with sigma=3 for process parameters variations, the Monte Carlo models of the technology has been used.

Best,

Authors.

Round 2

Reviewer 1 Report

This work can be accepted.